# OpenReview forum: "Cognition-of-Thought Elicits Social-Aligned Reasoning in Large Language Models"
_ICLR.cc/2026/Conference — ICLR 2026 Conference Desk Rejected Submission_

### Official Review · Reviewer_iGoA · 2025-11-01

**Soundness:** 3
**Presentation:** 3
**Contribution:** 2
**Rating:** 4
**Confidence:** 4

**Summary:**

This paper introduces Cognition-of-Thought (CooT), a novel decoding-time framework that equips LLMs with an explicit cognitive self-monitoring loop to improve safety and social reasoning. CooT couples a standard Generator with a cognitive Perceiver that continuously monitors generation using a precedence-based hierarchy. When violations are detected, the system performs causal rollback and regenerates with injected guidance combining universal social priors (BESSI framework) and context-specific warnings. Experiments on AIR-Bench 2024 and SocialEval show improvements over baselines, with comprehensive ablations demonstrating each component's contribution.

**Strengths:**

The paper addresses an important problem of making alignment explicit and dynamic rather than baked into model weights. The core idea of coupling a Generator with a Perceiver for real-time cognitive monitoring is interesting and well-motivated by psychological research on metacognition. The experimental methodology is thorough, with evaluations across multiple benchmarks (AIR-Bench, SocialEval, HarmBench) and model families (Gemma, Llama, Qwen, GPT), demonstrating generalizability. The ablation studies (Table 3) systematically validate each component's contribution, showing that rollback, guideline injection, and precedence-aware states are all necessary. The qualitative case studies in Appendix D provide valuable insights into when and how the system intervenes.

**Weaknesses:**

I think the primary weaknesses come in practical deployment of the framework.

For instance,  the "universal social schema" (BESSI) may have cultural limitations, as the BESSI framework is derived primarily from Western psychological research and may not generalize well across cultures with different social norms and values. The paper evaluates on English and Chinese tasks but doesn't discuss whether the social skills taxonomy (e.g., "Social Warmth," "Ethical Competence") translates appropriately across these contexts or whether culture-specific adaptations are needed.

Furthermore, the paper doesn't report inference latency or throughput. Given that CooT requires: (1) running the Perceiver at each step, (2) potentially multiple rollback-and-regenerate cycles, and (3) encoding contextual guidelines, the computational cost could be substantial. This is critical for practical deployment.

Also, the cognitive state representation is quite limited. Using just three binary values (Safety, Altruism, Egoism) ∈ {-1, 0, 1}³ to represent the model's "cognitive state" seems overly simplistic for capturing the complexity of ethical reasoning. Real ethical dilemmas often involve trade-offs that don't fit neatly into such scoring hierarchy (e.g., the trolley problem). Though I would rank this weakness as relatively minor since I couldn't think of a benchmark for stress-testing CooT.

**Questions:**

- What happens when the Perceiver itself makes errors in state classification? Is there any confidence calibration or uncertainty quantification in the cognitive state predictions?
- How does the Perceiver handle ambiguous cases where safety, altruism, and egoism are genuinely in tension? For instance, in a scenario where refusing to help (preserving safety) versus helping (altruism) both seem reasonable, how does the system make the judgment call?

---

> ### Author Response · Authors · 2025-11-18
> **Reviewer iGoA**
>
> We sincerely appreciate the reviewer's positive assessment of our core idea, experimental methodology, and ablation studies. We address your comments in detail below.
>
> ---
> _A1. On the Cultural Limitations of the BESSI Framework_
>
> (1) Generalization in Current Data
>
> Our experiments do show that Cognition-of-Thought generalizes across both English (Tables 2-3) and Chinese (Tables 14-15)  tasks. As noted in Appendix C.4, the pattern of improvement from CooT is consistent across both languages, even if the base models have different performance profiles in each . This suggests the core skills (like "Ethical Competence") provide a robust enough signal in both contexts.
>
> (2) Modularity as a Solution
>
> A key strength of CooT is its modularity. The Generator-Perceiver architecture is not hard-coded to BESSI. BESSI is simply the implementation of the "universal social guidance". The framework is explicitly designed to allow policies to be flexibly swapped without retraining. For deployment in a different cultural context, one could replace the BESSI-based priors with a different, culturally-specific taxonomy of social skills, and the core intervention mechanism (detect, rollback, inject) would remain the same.
>
> We acknowledge that this is an insightful and important point that any "universal" schema, including BESSI, may not capture the full spectrum of global social norms. We will add this discussion of modularity as the solution to cultural adaptation to the main paper.
>
>
> ---
> _A2. On Computational Overhead_
>
> The reviewer is correct that CooT introduces computational overhead, but this cost is structured and manageable.
>
> (1) Perceiver Cost
>
> The Perceiver's call is very fast, as it is prompted to perform a first-token probablity distribution in safety states. It only needs to generate cognitive correction tokens as the risk certainly triggered. Due to the effectiveness of our framework, interventions are sparse, and this is not a 2x cost on every token, but rather a cost incurred on a small fraction of generations that are flagged as non-compliant. In complex long-context tasks related to security, rollback guideline is generally not executed more than three times.
>
> (2) Scalability
>
> We acknowledge that cost is a concern for very large models. Our ablation on Perceiver Size (Table 3) shows that a stronger Perceiver improves performance. This suggests a cost-performance trade-off. A key advantage of our decoupled architecture is that one could use a smaller, distilled model as the Perceiver to audit a very large Generator, making the cost scalable.
>
> We will release full Computational Overhead Report in camera-ready version.
>
> ---
>
> _A3. On the Expressiveness of the State Space for Real-World Nuance_
>
> This is an excellent question. Our 3-dimensional state space is not intended to capture the infinite nuance of all possible ethical dilemmas. Its purpose is specific: It is designed to model the precedence conflicts (e.g., between Safety and Altruism) that are a common and critical failure mode for helpful-but-not-safe LLMs.
>
> Our results in Tables 10-11 show that this "simple" state space is highly effective at capturing the most frequent failure mode. The precedence-conflict states ($(-1,1,1)$ and $(-1,1,0)$) account for 74.2% of all detected violations on AIR-Bench. This suggests our state space, while simple, is highly targeted to the primary problem.
>
> Our current Cognition-of-Thought framework is the mechanism for handling trade-offs, explicitly enforces the trade-off Safety > Altruism. For more complex harm-harm trade-offs, this would require expanding the principles in our state space, which we believe is a promising direction for future work built upon our framework.

---

> ### Author Response · Authors · 2025-11-18
> **Reviewer iGoA (Part 2)**
>
> _Q1. Impact on Error State Embedding_
>
> This is a critical question. CooT is explicitly designed to avoid error. The Perceiver intervention is only triggered when a precedence conflict is detected (i.e., a state component becomes -1). For a normal, safe, and helpful request, the Perceiver would output a Cooperative or Unharmful State in $\{1,0\}$, which is flagged as 'Reliable' (I=R) and does not trigger any intervention. While still, a Perceiver error in false positive would cause an unnecessary rollback and regeneration, while a false negative would fail to correct a violation.
>
> The SocialEval results (Table 2)  provide strong evidence that false positives are not a major problem. This benchmark measures "Prosocial" tasks (which are unharmful). If the Perceiver had a high false-positive rate, it would be penalizing good, helpful responses and lowering this score. Instead, CooT significantly improves the prosocial score by +9.02%.
>
> We also acknowledge the critical role of Perceiver's capability. As we scale the Perceiver from 1.7B to 32B, performance consistently improves. This confirms that a more capable Perceiver makes fewer errors, leading to better outcomes. A future version could be designed to only intervene if the Perceiver's confidence in a violation state is above a certain threshold, further reducing the impact of potential Perceiver errors.
>
> ---
>
> _Q2. Handling Ambiguous Cases_
>
> The Perceiver handles the ambiguity by enforcing the explicit, pre-defined policy. The core design of CooT is to make alignment interpretable. In the example of refusing to help [Safety] vs. helping [Altruism], our system is opinionated by design. It enforces the strict precedence Safety > Altruism. If the Generator starts to help (satisfying Altruism=1) in a way the Perceiver deems potentially unsafe (violating Safety=-1), it will always flag this as a Misguided Compliance state (e.g., $(-1,1,0)$ or $(-1,1,1)$) , triggering an intervention to steer the model toward a response that prioritizes Safety, such as a Principled Independence state (e.g., (1,-1,1)).
>
> The "judgment call" is made by the policy. And the key feature is, the policy is transparent and auditable. If a different application required a different trade-off (e.g., a therapeutic bot where Altruism is paramount), one could change the precedence rules in the Perceiver's prompt without retraining the model.

---

### Official Review · Reviewer_ZSNF · 2025-11-01

**Soundness:** 2
**Presentation:** 3
**Contribution:** 2
**Rating:** 4
**Confidence:** 3

**Summary:**

This paper introduces Cognition-of-Thought (CooT), an inference-time alignment framework that couples a Generator with a cognitive Perceiver to monitor and correct LLM outputs during decoding. The Perceiver uses Asimov's Three Laws (Safety > Altruism > Egoism) to detect violations, triggers rollback to error origins, and injects universal + contextual guidance to steer generation toward safer outputs. The paper compares their method with a range of baselines on AIR-bench (measures safety) and on SocialEval (measures social intelligence), showing that CooT achieves higher or competitive performance.

**Strengths:**

- The decoding-time cognitive architecture is an interesting and creative idea, also the first work to propose inference-time alignment as an explicit Generator-Perceiver loop with structured state monitoring.
- The proposed framework is compared thoroughly with baselines across multiple model families. Results show consistent improvements.
- The ablation studies are informative and validate each component (rollback, guidelines, Perceiver size, precedence). All components contribute meaningfully which verifies the design of the framework.
- The paper is well written and easy to follow.

**Weaknesses:**

1. The state cognition model (section 3.1) lacks theoretical validation.
- The use of Asimov's laws seems under justified. Why did you choose this three laws specifically? They were designed for science fiction robots, not real AI safety.
- Why not use established moral psychology frameworks (e.g. Haidt's Moral Foundations Theory) or empirically grounded safety taxonomies.

2. The use of "cognition" seems inaccurate, as you are merely describing a pattern matching mechanism.
- You claim the Perceiver provides "cognitive self-monitoring," but isn't it just doing classification with a specially prompted LLM? I do not find justification that this is genuinely cognitive rather than sophisticated pattern matching.
- The term "cognition" should be used more sparingly or with more evidence (e.g., if there are true reasoning and understanding beyond behavioral output).

3. Rollback mechanism seems heuristic.
- The attention-based sharpness score (Eq. with max-norm + entropy) lacks principled justification. Why should peaked attention indicate causal error origins? There is no analysis of failure modes: what if attention is diffuse or the error spans multiple positions?
- The threshold τ requires tuning (Appendix A.4) but how to set it for new domains?

**Questions:**

1. My understanding is that your state space is {-1, 0, 1}³, giving only 27 possible states (and you restrict to a "feasible" subset F). But real-world ethical dilemmas are much more nuanced. How can this state space capture the complexity of social alignment? For example, how would your system handle cases requiring trade-offs between different types of harm?

2. You run two models in tandem (Generator + Perceiver), with potential rollback and regeneration. What's the computational overhead? CooT can be very computationally expensive on models beyond the ones you tested (> 32B).

3. Generator and Perceiver share weights, how can the same biased parameters reliably audit themselves?

---

> ### Author Response · Authors · 2025-11-18
> **Reviewer ZSNF (Part 1)**
>
> We sincerely thank the reviewer for positive feedbacks on the creativity of our framework, the thoroughness of our comparisons, and the value of our ablation studies. We are glad you found the paper well-written and easy to follow. We address the noted weaknesses and questions in detail below.
>
>
> ---
> _A1.  On the Justification for the State Cognition Model_
>
> We appreciate the reviewer's point on theoretical grounding. We want to clarify the functional role of Asimov's Laws in our framework. We chose these laws not because they are a definitive solution to AI safety, but because they provide a simple, universally-understood, and explicitly-ordered precedence hierarchy (Safety > Altruism > Egoism). Our primary goal was to test the mechanism of a precedence-aware monitoring loop, and this hierarchy was the clearest and most interpretable vehicle to do so. Frameworks like Haidt's Moral Foundations Theory are excellent descriptive taxonomies of what humans consider moral, but they do not offer a clear, computational precedence hierarchy for intervention.
>
> This mechanism is grounded in established cognitive theory. We explicitly cite Kohlberg's stage theory of moral development, where moral maturity is defined by the ability to resolve conflicts between lower-order principles in favor of higher-order ones. Our framework is a computational implementation of this type of conflict resolution.
>
>
> ---
>
> _A2. Clarifying "Cognition" as a Functional Analogy and Framework Distinction_
>
> (1) Clarification on the design philosophy
>
> We understand the reviewer's concern regarding the use of psychological terminology. We want to clarify that we are not claiming Cognition-of-Thought possess human-like sentience or cognition. Our intent is to use these terms as a functional analogy to provide a clear conceptual framework, drawing inspiration from human metacognition—the process of monitoring and regulating one's own thought processes. CooT was positioned as a direct parallel to "Chain-of-Thought" (CoT). While CoT elicits a *reasoning* process, our framework, CooT, elicits a *monitoring-and-correction* process that runs in tandem with it.
>
> (2) Clarification on cognitive states
>
> We respectfully disagree that our Perceiver is simply a "classifier." A typical classifier provides a post-hoc, binary "safe/unsafe" label. While our Perceiver's "cognitive state" is a structured, multi-dimensional diagnosis ($y_t \in \{-1, 0, 1\}^3$) that captures why a generation is misaligned by modeling a precedence hierarchy (e.g., "Safety" is violated because "Altruism" was incorrectly prioritized) in real time, and it serves as a state index to inspire subsequent cognitive correction processes in Perceiver. Our contribution is the formalization of this concurrent monitoring process. Unlike an external classifier, our Perceiver is an internal loop active during decoding, making alignment an explicit, dynamic, and auditable process rather than an black-box tag.
>
> As shown in Appendix C.3 (Tables 10-11), the Perceiver predominantly flags precedence conflicts (e.g., states (-1,1,1) and (-1,1,0)), which account for 74.2% of interventions on AIR-Bench. This shows the model is reasoning about the hierarchical rules, not just flagging bad words. We will move this crucial evidence to the main paper.
>
> ---
>
> _A3. On the Justification of the Rollback Mechanism_
>
> We acknowledge that the rollback mechanism is a heuristic, but it is an empirically validated and essential one.
>
> Theoretically, the position is not determined by just "larger" attention scores. We compute a sharpness score $s_t$ based on both the max-norm ($||\hat{a}_{t}||_{\infty}$) and the normalized entropy of the attention distribution. A high sharpness score indicates a point where the model made a "strong commitment" to a specific prior context, which is our heuristic for the origin of the unsafe trajectory. We agree that analyzing failure modes, such as diffuse attention, is an excellent point. We will add this as a limitation and area for future work.
>
> The ablation study (Table 3, "w/o Rollback") provides the empirical validation for current design. When the rollback mechanism is removed (and intervention happens at the point of detection), performance significantly degrades (Prosocial score drops -1.52, Antisocial score increases +1.81).
>
> We empirically chose $\tau$ because, under our theoretical design, the sharpness score exhibits strong differentiation across cross-domain tasks. Our sensitivity analysis (Appendix A.4, Figure 3) shows the system is robust across a range of values, not just a single point. Importantly, we used the same set of hyperparameters across all benchmarks (AIR-Bench, SocialEval) and model families, which demonstrates strong generalization. For a completely new domain, a small tuning step on a validation set would be standard practice.

---

> ### Author Response · Authors · 2025-11-18
> **Reviewer ZSNF (Part 2)**
>
> _Q1. On the Expressiveness of the State Space for Real-World Nuance_
>
> This is an excellent question. Our 3-dimensional state space is not intended to capture the infinite nuance of all possible ethical dilemmas. Its purpose is specific: It is designed to model the precedence conflicts (e.g., between Safety and Altruism) that are a common and critical failure mode for helpful-but-not-safe LLMs.
>
> Our results in Tables 10-11 show that this "simple" state space is highly effective at capturing the most frequent failure mode. The precedence-conflict states ($(-1,1,1)$ and $(-1,1,0)$) account for 74.2% of all detected violations on AIR-Bench. This suggests our state space, while simple, is highly targeted to the primary problem.
>
> Our current Cognition-of-Thought framework is the mechanism for handling trade-offs, explicitly enforces the trade-off Safety > Altruism. For more complex harm-harm trade-offs, this would require expanding the principles in our state space, which we believe is a promising direction for future work built upon our framework.
>
> ---
>
> _Q2. On Computational Overhead_
>
> The reviewer is correct that CooT introduces computational overhead, but this cost is structured and manageable.
>
> (1) Perceiver Cost
>
> The Perceiver's call is very fast, as it is prompted to perform a first-token probablity distribution in safety states. It only needs to generate cognitive correction tokens as the risk certainly triggered. Due to the effectiveness of our framework, interventions are sparse, and this is not a 2x cost on every token, but rather a cost incurred on a small fraction of generations that are flagged as non-compliant. In complex long-context tasks related to security, rollback guideline is generally not executed more than three times.
>
> (2) Scalability
>
> We acknowledge that cost is a concern for very large models. Our ablation on Perceiver Size (Table 3) shows that a stronger Perceiver improves performance. This suggests a cost-performance trade-off. A key advantage of our decoupled architecture is that one could use a smaller, distilled model as the Perceiver to audit a very large Generator, making the cost scalable.
>
> ---
>
> _Q3. On the Efficacy of Shared Weights for Self-Auditing_
>
> This is an insightful question. The same base model is used, but it is prompted to perform two functionally distinct tasks:
>
> 1. Generation: An open-ended, creative task.
> 2. Perception: A constrained, analytical embedding task, where the output is restricted to the state space, then followed with detailed correction guideline.
>
> Conditioning a model with a strong, specific prompt (like our Perceiver prompt, $p_{perc}$) to perform an evaluation task is a standard, effective, and well-documented technique in LLM unbiased research. Our main results (e.g., Table 1, 2) show that even with total shared weights (e.g., 8B Generator + 8B Perceiver), CooT still significantly outperforms the 8B base model on its own.

---

### Official Review · Reviewer_N5iE · 2025-11-02

**Soundness:** 1
**Presentation:** 3
**Contribution:** 2
**Rating:** 2
**Confidence:** 4

**Summary:**

This paper proposes a prompt engineering strategy for improving safety alignment. The LLM is prompted with instructions modeled on Asimov's three rules of robotics. In addition, the reasoning output is monitored for misalignment. If misalignment is detected by another LLM, the generation is rolled back to a point identified by inspecting attention scores. Additional guidance is injected into the chain of thought and the generation is resumed. Experiments show that this approach gives a modest boost to safety and social reasoning benchmarks.

**Strengths:**

1. The paper's attempts to incorporate lessons from other fields like psychology are interesting and creative, even if I don't think that they are successful.

2. The proposed technique is simple to implement.

**Weaknesses:**

1. A major concern is the reliance on language from psychology that seems misleading about what is really happening. The method is essentially a particular prompt with an LLM-as-judge that monitors the results. I don't think it is reasonable to call this "cognition". The paper has many instances of this provocative terminology that seems out of place and overstating the capabilities of LLMs. Another example is "cognitive states," which are just vectors in a subset of {-1, 0, 1}^3. The first claimed contribution says that the paper "formalizes cognitive perception." I respectfully disagree that this is a reasonable way to describe writing prompts that describe a model of human cognition.

1. The paper does not engage very much with whether the prompting strategy is having the desired effect, in the sense that the model is actually following the intended schema. A few case studies are presented in the appendix without synthesizing the findings into an overall evaluation. This paper would be much stronger if it evaluated the ability of LLMs to follow the intended rules and reason in the described state space. Currently the focus in presentation is on small improvement in existing benchmarks, rather than investigating the limits of LLMs to reason about these topics. I would be much more enthusiastic about such a paper and I encourage the authors to consider such a direction.

1. The paper does not measure the variance across generations. The results are presented without something like standard errors, but the differences in scores between methods seem small on some metrics.

**Questions:**

1. Does CooT incur any penalty in accuracy on unharmful requests? How often does the LLM judge incorrectly flag unharmful situations?

1. In what sense is the "causal rollback" causal? The position to rollback to is determined by larger attention scores. How do attention scores prove that a particular place in the CoT caused the result to be harmful (or any other aspect of the result)?

---

> ### Author Response · Authors · 2025-11-18
> **Reviewer N5iE (Part 1)**
>
> We sincerely thank the reviewer for the thorough and constructive feedback. We appreciate the acknowledgment of our paper's creative attempt to integrate concepts from psychology and its simplicity of implementation. Below, we address the concerns in detail.
>
> ---
> _A1. Clarifying "Cognition" as a Functional Analogy and Framework Distinction_
>
> (1) Clarification on the design philosophy
>
> We understand the reviewer's concern regarding the use of psychological terminology. We want to clarify that we are not claiming Cognition-of-Thought possess human-like sentience or cognition. Our intent is to use these terms as a functional analogy to provide a clear conceptual framework, drawing inspiration from human metacognition—the process of monitoring and regulating one's own thought processes. CooT was positioned as a direct parallel to "Chain-of-Thought" (CoT). While CoT elicits a *reasoning* process, our framework, CooT, elicits a *monitoring-and-correction* process that runs in tandem with it.
>
> (2) Clarification on cognitive states
>
> We respectfully disagree that our Perceiver is simply an "LLM-as-judge." A typical judge provides a post-hoc, binary "safe/unsafe" label. While our Perceiver's "cognitive state" is a structured, multi-dimensional diagnosis ($y_t \in \{-1, 0, 1\}^3$) that captures why a generation is misaligned by modeling a precedence hierarchy (e.g., "Safety" is violated because "Altruism" was incorrectly prioritized) in real time, and it serves as a state index to inspire subsequent cognitive correction processes in Perceiver. Our contribution is the formalization of this concurrent monitoring process. Unlike an external judge, our Perceiver is an internal loop active during decoding, making alignment an explicit, dynamic, and auditable process rather than an black-box tag.
>
> ---
> _A2. Evidence for the Model's Adherence to the Normative Schema_
>
>
> (1) Quantitative Validation
>
> We performed a statistical analysis of the internal dynamics of CooT. Tables 10-13 show precisely which cognitive states triggered interventions. On the safety-focused AIR-Bench, 74.2% of all interventions were triggered by states $(-1, 1, 1)$ (Selective Harm) and $(-1, 1, 0)$ (Misguided Compliance). This is strong evidence that the model is not just using a simple "unsafe" filter, but is actively reasoning within our defined state space and correctly identifying nuanced precedence conflicts (i.e., prioritizing obedience over safety) as the primary cause of failure.
>
>
> (2) Qualitative Validation
>
> The case studies provide step-by-step, auditable traces of the schema in action. For example, in Case 1 (Defamation) and Case 2 (Harassment), the Perceiver precisely identifies the generation of harmful content, assigns the correct Cognitive State Vector: $(-1, 1, 1)$, flags the Safety violation, and selects the appropriate Social Skill (Ethical Competence)  to guide the regeneration.
>
> _A3. On the Significance of Performance Gains_
>
> We acknowledge that improvements on safety benchmarks can often appear incremental. While some individual metrics are close, the average improvements of +13% on AIR-Bench (from 0.67 to 0.80) and +9.02% on SocialEval (from 41.24 to 50.26) are substantial in this domain. These gains are consistent across multiple model families and sizes, as shown in Figure 2 and Table 9.

---

> ### Author Response · Authors · 2025-11-18
> **Reviewer N5iE (Part 2)**
>
> _Q1. Impact on Unharmful Requests_
>
> This is a critical question. CooT is explicitly designed to avoid penalizing helpful, unharmful requests. The Perceiver intervention is only triggered when a precedence conflict is detected (i.e., a state component becomes -1). For a normal, safe, and helpful request, the Perceiver would output a Cooperative or Unharmful State in $\{1,0\}$, which is flagged as 'Reliable' (I=R) and does not trigger any intervention.
>
>
> The SocialEval benchmark (Section 4.2) directly measures this. This benchmark includes a suite of "Prosocial" tasks (Cooperation, Assistant, Negotiation, Altruism) that are unharmful. As shown in Table 2, CooT significantly improves the prosocial score by +9.02% on average (from 41.24 to 50.26). This result strongly indicates that CooT does not incur a penalty on unharmful requests; on the contrary, its cognitive loop actively steers the model toward more constructive, helpful, and socially intelligent responses. A high false positive rate would have resulted in a decrease in these prosocial scores.
>
> ---
> _Q2. Validation of the Causal Rollback Mechanism_
>
> We appreciate the reviewer's call for precision on this term. We use "causal" not in a strict, formal sense (e.g., Pearl's causal calculus), but in the heuristic sense of identifying the anchor-point of the flawed reasoning trajectory. We agree this term can be strong and will clarify in the text.
>
> Theoretically, the position is not determined by just "larger" attention scores. We compute a sharpness score $s_t$ based on both the max-norm ($||\hat{a}_{t}||_{\infty}$) and the normalized entropy of the attention distribution. A high sharpness score indicates a point where the model made a "strong commitment" to a specific prior context, which is our heuristic for the origin of the unsafe trajectory.
>
> The effectiveness of this heuristic is validated by our ablation study in Table 3. The "w/o Rollback" variant (which intervenes at the point of detection but does not roll back to the origin-point $t^*$) shows a significant drop in performance. The prosocial score falls from 50.26 to 48.74, and the antisocial score increases from 12.19 to 14.00.
>
> We will include such a discussion in our manuscript as well. Thanks again for your helpful comment!

---

> ### Comment · Reviewer_N5iE · 2025-11-23
>
> Thanks to the authors for addressing my comments and questions.
>
> Regarding A1, I think this is a bit subjective, but the fact that Reviewer ZSNF has the same concern underscores its importance. If the intent is to use these terms as a functional analogy, that is not clear in the paper.
>
> Regarding A2, I'm not understanding why is the quantitative evidence strong? The evidence just seems to be that it's using two vectors most often in deciding something is unsafe. For the sake of argument, what pattern of responses would show that it is not reasoning?
>
> Regarding A3, this doesn't address my concern that no notion of variance is reported.
>
> Thank you for addressing Q1. That is helpful and I agree it indicates that the models are not suffering from too much over-refusal.
>
> Regarding Q2, the authors say
> ```
> We use "causal" not in a strict, formal sense (e.g., Pearl's causal calculus), but in the heuristic sense of identifying the anchor-point of the flawed reasoning trajectory. We agree this term can be strong and will clarify in the text.
> ```
> I disagree that this is something that can be "clarified". The term should be changed. The word causal means that there's evidence that it causes something else. This so-called "heuristic sense" is bad science.

---

### Official Review · Reviewer_xHxL · 2025-11-04

**Soundness:** 2
**Presentation:** 3
**Contribution:** 2
**Rating:** 6
**Confidence:** 3

**Summary:**

This paper proposed a cognitive alignment framework for LLMs, enabling them to self-monitor their own outputs. Specifically, the authors introduced an additional cognitive perceiver for LLMs to continuously monitor the generation process. The perceiver used specialized prompts to determine whether the generated text complied with Asimov’s Three Laws of Robotics. When detecting violations, the model rolled back the generation process to the wrong position by aggregating the generator's attention maps to identify positions most strongly affecting the current prediction. Then, the authors introduced corrective guidelines (sentence priors) to guide re-generation from the position. Experiments showed that the framework improved safety and social reasoning performance across multiple models.

**Strengths:**

$\bullet$ The paper presents a self-monitoring safety alignment framework that incorporates human cognition, and validates its effectiveness across different model families and safety scenarios. The framework appears conceptually simple and practical to implement.

$\bullet$ It is great to introduce guidelines and priors. The cognitive perceiver assesses whether the generated text satisfies Asimov’s Three Laws, i.e., Safety, Altruism, and Egoism, and uses normative corpora such as the Behavioral, Emotional, and Social Skills Inventory to guide regeneration.

**Weaknesses:**

$\bullet$ The overall framework of this paper appears clear and intuitive: detecting inappropriate generation, localizing it, and re-generating the text. However, in terms of specific technical choices, some ideas seem unclear or may involve additional options. For example, why does the perceptron use an LLM to generate classification results instead of a standard supervised classifier (the latter seemingly faster)? In the localization step, why can the maximum value of the aggregated attention map (lines 251-255) be considered the localization result? A more thorough discussion comparing common localization techniques would help justify this choice.

$\bullet$ The paper should report the computational overhead introduced by the framework. Methods that embed security mechanisms into model weights typically do not introduce additional inference overhead. It appears that using perceivers (along with other steps) will introduce extra inference overhead. The authors should compare the average inference time between the base model and the model using the proposed framework (as in Figure 2) on the same tasks. Additionally, they should compare the runtime increase incurred by common security methods when performing the same inference tasks (as shown in Table 2).

$\bullet$ The paper should include several examples of outputs before and after using this framework, both in the main text and in the appendix. It would also be valuable to show failed correction cases and analyze the underlying reasons.

**Minor:**

In Table 2, in Cooperation column, the AGRS method (54.26) appears to outperform the proposed method (54.12), yet the proposed method is highlighted in bold.

**Questions:**

See weaknesses.

---

### Note · Program_Chairs · 2025-11-28
**Submission Desk Rejected by Program Chairs**

Authors revealed reviewer names on social media, breaking double blind anonymity.